# Multimodal Imaging Characteristics in Unilateral Occlusive Macular Telangiectasia with Atypical X-Shaped Lesion

**DOI:** 10.3390/diagnostics15060754

**Published:** 2025-03-17

**Authors:** Abdullah Ağın, Ilknur Turk, Burcu Yakut

**Affiliations:** Department of Ophthalmology, University of Health Science, Haseki Training and Research Hospital, Istanbul 34130, Turkey; ilknurturk99@gmail.com (I.T.); burcuykt@hotmail.com (B.Y.)

**Keywords:** macular telangiectasia, multimodal imaging, fundus fluorescein angiography, fundus autofluorescence, optical coherence tomography, optical coherence tomography angiography

## Abstract

Macular Telangiectasia (MacTel) is a rare retinal vascular disorder, with Type 3a MacTel being a distinct form characterized by retinal ischemia with the classical findings of MacTel, such as juxtafoveal telangiectasis, right-angled venules, and deep capillary plexus involvement without central nervous system findings. This case presents a novel X-shaped lesion pattern and ischemic features, expanding the known imaging spectrum of MacTel. A 53-year-old male with diabetes and a history of aripiprazole use presented with persistent blurred vision, a black curtain sensation, and metamorphopsia in the right eye. Visual acuity was 0.8 in the right eye and 1.0 in the left. A multimodal imaging approach, including fundus photography, fundus autofluorescence (FAF), fluorescein angiography (FFA), optical coherence tomography (OCT), and optical coherence tomography angiography (OCTA), was used to evaluate structural and vascular abnormalities. Fundus examination revealed an X-shaped hypopigmented lesion with central pigmentation. FAF showed hypoautofluorescence, indicating chronic RPE loss, and no loss of foveal autofluorescence was observed. FFA demonstrated progressive hyperfluorescence with perifoveal aneurysmal and telangiectatic vessels, along with a slightly enlarged foveal avascular zone (FAZ), suggesting ischemic involvement. OCT revealed intraretinal cysts, a disruption of the ellipsoid zone and external limiting membrane, pigment epithelial detachment, and increased choroidal backscattering. OCTA confirmed right-angled venules, aneurysmal telangiectatic vessels, and localized ischemia predominantly affecting the deep capillary plexus. This case highlights a rare variant of Type 3a MacTel with a unique X-shaped lesion. The presence of juxtafoveal telangiectasis, vascular occlusion, right-angled venules, and deep capillary plexus changes supports the diagnosis. Multimodal imaging played a critical role in characterizing the disease and differentiating it from other macular disorders, contributing to an expanded understanding of the clinical and imaging spectrum of MacTel.

**Figure 1 diagnostics-15-00754-f001:**
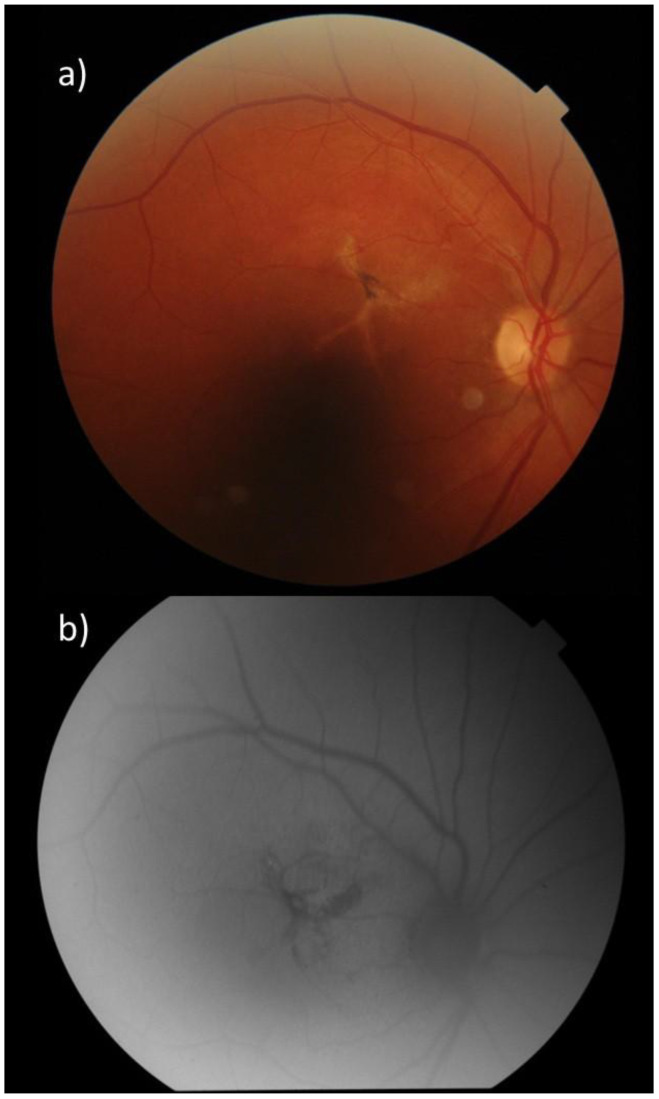
(**a**) In the fundus photograph (FF), an X-shaped hypopigmented area with a central cluster of pigmented spots is observed superonasal to the fovea. (**b**) In fundus autofluorescence (FAF), hypoautofluorescence corresponding to the lesion area is detected, attributed to retinal pigment epithelium loss associated with chronicity, and no loss of foveal autofluorescence is observed, which is quite typical and occurs secondary to foveal pigment loss in MacTel [1,2].

**Figure 2 diagnostics-15-00754-f002:**
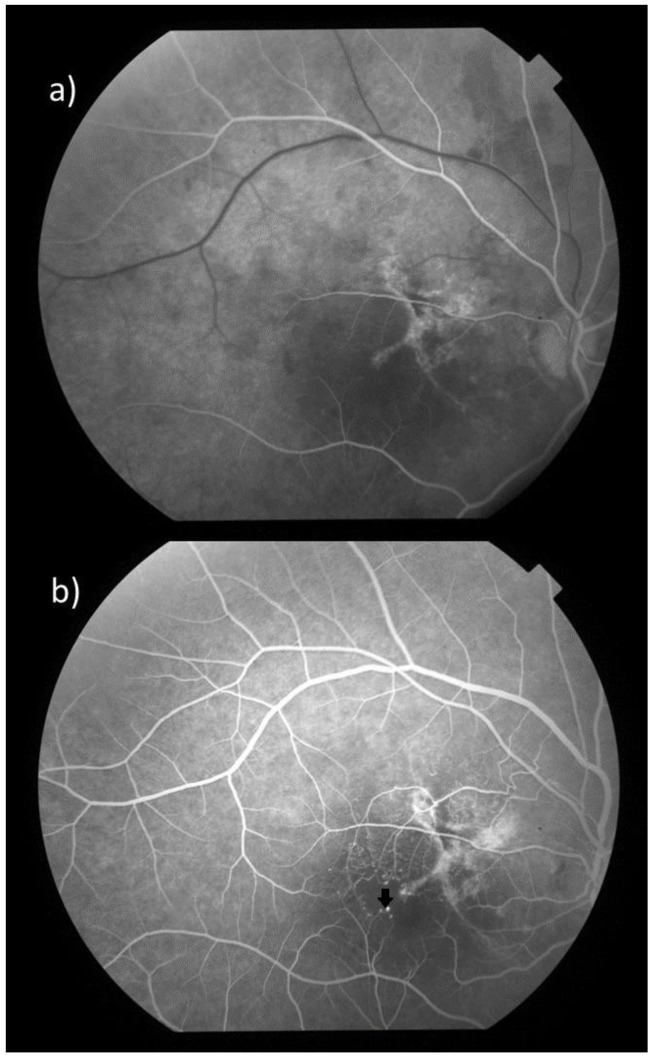
In Fundus Fluorescein angiography (FFA), the X-shaped lesion demonstrates progressive hyperfluorescence from the (**a**) early arteriovenous phase (captured at 15 s post-injection) to (**b**) late phases (captured at 5 min post-injection). Additionally, perifoveal aneurismal and telangiectatic vessels temporal and superior to the macula and right-angled venules (arrow) are observed in all phases. Also, a slightly enlarged foveal avascular zone (FAZ) is observed.

**Figure 3 diagnostics-15-00754-f003:**
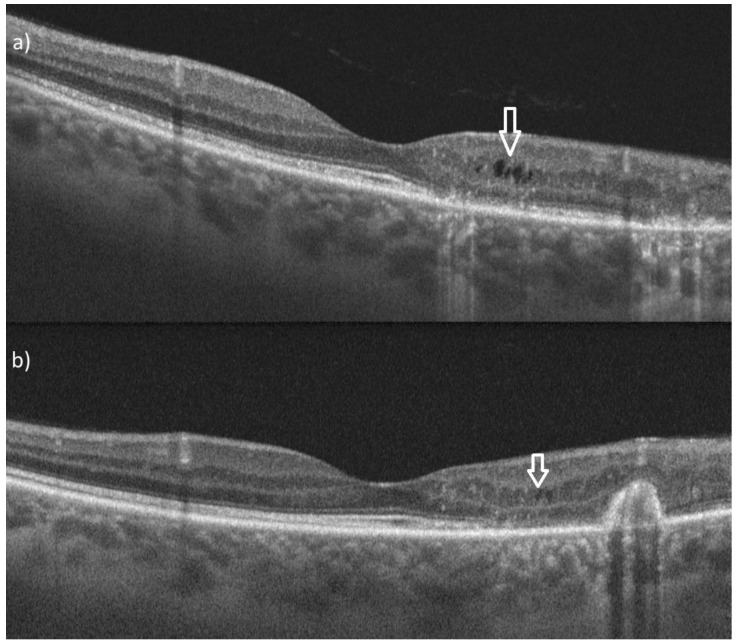
(**a**,**b**) In optical coherence tomography (OCT), structural damage is noted at the lesion site, involving the retinal pigment epithelium, ellipsoid zone, and external limiting membrane. Increased choroidal backscattering and a small pigment epithelial detachment containing hyper-reflective material and intraretinal cysts are also observed (white arrow).

**Figure 4 diagnostics-15-00754-f004:**
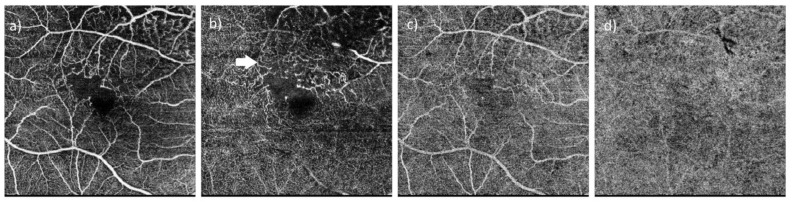
In optical coherence tomography angiography (OCTA), aneurismatic telangiectatic vessels and ischemia in the superior-temporal region of the fovea are especially evident in the deep capillary plexus, and right-angled venules (white arrow) are also shown. (**a**) Superficial capillary plexus layer, (**b**) deep capillary plexus, (**c**) outer retina, and (**d**) choriocapillaris.

## Data Availability

No new data were created or analyzed in this study.

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
