# Peer review of "Multimodal Imaging Characteristics in Unilateral Occlusive Macular Telangiectasia with Atypical X-Shaped Lesion"

_diagnostics, 2025, doi:10.3390/diagnostics15060754_

Round 1
Reviewer 1 Report
Comments and Suggestions for Authors
The authors are requested to revise with the following changes:
- What observations led to the identification of the particular disease?
- Proper introduction is missing in context to what Type 2 macular Telangiectasia mean.
- And a brief conclusion of all the findings that led to diagnosis of the disease?
- Figure 2- What's the duration of early and late phase after diagnosis ?
- Figure 3-arrow missing for intraretinal cysts.
- Figure 4- Arrows are missing as mentioned in the legend.
Author Response
We sincerely appreciate your valuable feedback and constructive comments regarding our manuscript. Below, we have addressed each of your concerns and made the necessary revisions accordingly. All changes have been incorporated into the manuscript, and we have highlighted the specific modifications.
1. What observations led to the identification of the particular disease?
Response: The diagnosis was based on multimodal imaging findings, including fundus photography, fundus autofluorescence (FAF), fluorescein angiography (FFA), optical coherence tomography (OCT), and optical coherence tomography angiography (OCTA). Key observations included:
An X-shaped hypopigmented lesion with central pigmentation on fundus examination.
FAF showing hypoautofluorescence due to retinal pigment epithelium (RPE) loss.
FFA revealing progressive hyperfluorescence with perifoveal aneurysmal and telangiectatic vessels.
OCT detecting significant structural damage, including intraretinal cysts, pigment epithelial detachment, and increased choroidal backscattering.
OCTA demonstrating right-angled venules, aneurysmal telangiectatic vessels, and localized ischemia.
Additionally, a slightly enlarged foveal avascular zone (FAZ) was observed in FFA, supporting the ischemic nature of the disease.
2. Proper introduction is missing in the context of what Type 2 Macular Telangiectasia means.
Response: We have revised the introduction to introduce Type 3a Macular Telangiectasia (MacTel) instead of Type 2 because of ischemic area in deep capillary plexus and slightly enlarged foveal avascular zone. This correction ensures that the description aligns with the classification and highlights the distinguishing ischemic features of Type 3a MacTel. While Müller cell damage is primarily considered the underlying mechanism in Type 2 MacTel, Type 3a MacTel is extremely rare and distinguished from other MacTel subtypes by the presence of ischemia.
3. A brief conclusion of all the findings that led to the diagnosis of the disease?
Response: In addition to the classical findings of MacTel, the absence of trauma history, no loss of foveal autofluorescence in FAF, the presence of juxtafoveal telangiectasia and right-angled venules, as well as ischemia in the deep capillary plexus, which could be associated with occlusive vasculopathy, led us to consider MacTel as the most likely diagnosis.
4. Figure 2 - What’s the duration of early and late phase after diagnosis?
Response: The early arteriovenous phase was captured at 1 minute, while the late phase was recorded at 5 minutes post-injection. This clarification has been added to the manuscript.
5. Figure 3 - Arrow missing for intraretinal cysts.
Response: We have added an arrow to indicate the intraretinal cysts in Figure 3 to ensure clarity.
6. Figure 4 - Arrows are missing as mentioned in the legend.
Response: We have updated Figure 4 by adding arrows to highlight aneurysmatic telangiectatic vessels and ischemia in the deep capillary plexus, as stated in the legend.
We appreciate your time and effort in reviewing our manuscript. We believe these revisions have significantly improved the clarity and accuracy of our work.
Reviewer 2 Report
Comments and Suggestions for Authors
This is a very interesting case of type 2 Macular Telangiectasia. The multimodal imaging presentation is thorough, the image quality is excellent and the diagnosis is clear. This paper can be clinically significant, as it may help a clinician reach a MacTel 2 diagnosis. This paper describes an unusual presentation of MacTel 2 and it provides all the necessary multimodal imaging to better present it. In terms of originality, this is the first presentation of a MacTel 2 case with this appearance in fundoscopy. This is helpful to clinicians, as it highlights an uncommon fundoscopic presentation of this otherwise common condition.
I have no concerns regarding the methodology ( the Interesting Images category is rather simple in terms of preparation).
I have no further concerns regarding this paper; I think it is interesting, appropriately prepared and useful.
Author Response
Thank you so much for your detailed evaluation. Best regards
Round 2
Reviewer 1 Report
Comments and Suggestions for Authors
1. Figure 2- I wanted to know the time requires for the disease to progress into late phase after diagnosis.
2. And the duration of time (in months or days) between the early and late phase.
3. How will the author denote a disease is in early phase and late phase?
Author Response
1. Figure 2 - I wanted to know the time requires for the disease to progress into late phase after diagnosis.
Response: The disease was incidentally detected during the examination, and there is no follow-up data available to determine the time required for progression to the late phase.
2. And the duration of time (in months or days) between the early and late phase.
Response: There is no specific timeline provided in the study for the transition between early and late phases. The progression rate varies among individuals, and there is no established time frame for this change, as I mentioned in the answer of Figure 1. We don't know the progression of the disease in this patient.
3. How will the author denote a disease is in early phase and late phase?
Response: Thank you for your understanding. In the nomenclature, MacTel is classified as non-proliferative and proliferative when complicated by choroidal neovascular membrane (CNVM). However, there is no universal standardization defining early or late phases in MacTel, so we did not include such a classification in our manuscript. If you'd like, we can also incorporate this information into the manuscript, but we'll leave the decision to you. We appreciate your time in reviewing our manuscript and your valuable contributions.